



**A Predictive Algorithm For Wetlands In Deep Time Paleoclimate Models**
David J. Wilton[1], Marcus Badger[2,3,4], Euripides P. Kantzas[1], Richard D. Pancost[3], Paul J.
Valdes[4], David J. Beerling[1]
[1]Dept. Animal and Plant Sciences, The University of Sheffield, Sheffield, S10 2TN, UK
[2]School of Environment, Earth and Ecosystem Sciences, The Open University, Milton
Keynes, MK7 6AA
[3]Organic Geochemistry Unit, The Cabot Institute, School of Chemistry, School of Earth
Sciences, The University of Bristol, Bristol, BS8 1TH, UK
[4]Bristol Research Initiative for the Dynamic Global Environment (BRIDGE), The Cabot
Institute, School of Geographical Sciences, The University of Bristol, BS8 1TH, UK
*Correspondence to*: David J. Wilton (d.j.wilton@shef.ac.uk)
**Abstract.** Methane is a powerful greenhouse gas produced in wetland environments via
microbial action in anaerobic conditions. If the location and extent of wetlands are unknown,
such as for the Earth many millions of years in the past, a model of wetland fraction is
required in order to calculate methane emissions and thus help reduce uncertainty in the
understanding of past warm greenhouse climates. Here we present an algorithm for predicting
inundated wetland fraction for use in calculating wetland methane emission fluxes in deep
time paleoclimate simulations. The algorithm determines, for each grid cell in a given
paleoclimate simulation, the wetland fraction  predicted by a nearest neighbours search of
modern day data in a space described by a set of environmental, climate and vegetation
variables. To explore this approach, we first test it for a modern day climate with variables
obtained from observations and then for an Eocene climate with variables derived from a
fully coupled global climate model (HadCM3BL-M2.2). Two independent dynamic
vegetation models were used to provide two sets of equivalent vegetation variables which
yielded two different wetland predictions. As a first test the method, using both vegetation
models, satisfactorily reproduces modern data wetland fraction at a course grid resolution,
similar to those used in paleoclimate simulations. We then applied the method to an early
Eocene climate, testing its outputs against the locations of Eocene coal deposits. We predict
global mean monthly wetland fraction area for the early Eocene of 8 to $10 \times 10^6$ km$^2$ with
corresponding total annual methane flux of 656 to 909 Tg CH$_4$ year$^{-1}$, depending on which of
two different dynamic global vegetation models are used to model wetland fraction  and
methane emission rates. Both values are significantly higher than estimates for the modern-
day of $4 \times 10^6$ km$^2$ and around 190 Tg CH$_4$ year$^{-1}$ (Poulter et. al. 2017, Melton et. al., 2013).

## 1 Introduction

Methane (CH$_4$) is a powerful greenhouse gas. As well as absorbing infrared radiation from
the Earth's surface it also contributes to additional indirect warming through its





photochemistry and oxidation to $CO_2$ in the atmosphere (IPCC 2013). Therefore, Earth
system models used to reconstruct ancient climate or develop future climate scenarios must
either assume atmospheric methane concentrations as a boundary condition and/or
incorporate dynamic methane fluxes from natural sources (Beerling et al. 2011). The main
natural source of methane is wetland environments via microbial action in anaerobic
conditions (Whiticar, 1999), but methane fluxes from wetlands are also modulated by
climatic factors such as temperature (Westermann, 1992). Therefore, in order to model fluxes
of methane to the atmosphere both the extent and locations of wetlands need to be known.
For modern day, recent past and near future scenarios, maps of observed wetland extent
(Prigent et al. 2007, Papa et al. 2010, Schroeder et al., 2015, Poulter et al, 2017) can be used
or wetland extent can be calculated at a sub-grid level from fine resolution topographical data
(as in the TOPMODEL approach of Beven and Kirkby (1979), Lu and Zhuang (2012),
Stocker et al. (2014), Lu et al. (2016)), as wetlands only form where the ground is relatively
flat.
For the study of deep time paleoclimates (many millions of years in the past) there are no
direct observations of wetland extent, and the topography is only known on relatively coarse
resolutions of around 0.5 ° at best. Therefore, any model calculation of wetland extent must
either rely on using approximate knowledge of the topography or not rely on the topography
at all. Previous studies (Beerling et al., 2011, Valdes et al., 2005) classified grid cells as either
producing or not producing methane, based on either: i) a month being within a defined melt
season, for grid cells where mean monthly temperature drops below 0 °C at some point in the
year; or ii) precipitation being greater than evapotranspiration. They then scaled emissions by
empirically derived functions of the variance or standard deviation of orography, at the best
resolution available. The scaling effectively reduces methane emission rates in grid cells
where elevation varies significantly and are therefore unlikely to have substantial wetlands
within them, but relies on what may be quite coarse resolution topography not able to resolve
sub-grid scale variations.
In this work we develop a nearest neighbour-based algorithm to predict the fraction of a
specified area that is wetland (FW). We base this on modern day reference data set of  FW
and corresponding environmental variables, empirically associating the FW observations with
corresponding observed climate data and vegetation data calculated using one of two
dynamic global vegetation models (DGVMs).  We demonstrate its application by predicting
FW and $CH_4$ fluxes for an early Eocene (52 Ma) model climate, an interval of greenhouse
warming (Zachos et al., 2008) when sedimentary records indicate the existence of large areas
of wetlands (Sloan et al., 1992, Beerling et al., 2009). For the Eocene, the same climate
variables are obtained from a fully coupled global climate model and vegetation variables are
derived from the same DGVMs.  We then predict FW for the Eocene by analysis and
comparison to the modern-day reference data. We note that different reference sets,
vegetation models or climate models will likely yield different results and these should be
explored in future work, but our aim here is to demonstrate this approach and its potential
rather than to produce a model-model intercomparson.
Firstly, we describe modern day wetland data  at 0.5° spatial resolution and a monthly time
step for a mean modern day year, along with climate and vegetation data which we later use
as a reference data set. We then describe two test data sets at lower spatial resolution,



equivalent to that used in paleoclimate models, again for a single year. The first of these is for
the modern day and derived by interpolation of the reference data and the second is derived
from a paleoclimate model of the early Eocene. We briefly describe unsuccessful attempts to
model FW before moving on to the Nearest Neighbours method we found to be successful.
We also describe the model used to calculate wetland methane emissions. We then discuss
the model results for the modern day test data set and then Early Eocene climate. For the
modern day test data set the nearest neighbour method should yield strong agreement, since it
is simply a downscaled version of the reference data; these results, therefore, serve to
demonstrate whether or not a generalised form of the method can be successfully applied to
prediction of FW for a climate very different to the modern day. We then apply this method
to prediction of FW for the Eocene, and show that we can tune it by using the locations of
coal deposits as wetland proxies.

**2 Data and Methods**
**2.1 Modern day reference data**
We use a modern-day reference data set of observed FW with corresponding environmental
data to develop an algorithm for the prediction of FW in the past, i.e. we assume that there
exists a relationship between FW and the environmental variables compiled in the reference
data and then apply that relationship to predicting FW in the past. We use the recently
developed SWAMPS-GLWD (Poulter et al., 2017), which improves on the Surface Water
Microwave Product Series (SWAMPS) (Schroeder et al., 2015) by adding Global Lakes and
Wetlands Database (GLWD) (Lehner and Doll 2004) data, correcting the SWAMPS dataset
in regions where this satellite derived dataset fails to detect water beneath closed canopies.
We calculated the average monthly FW at each $0.5° \times 0.5°$ grid cell for the years 2000 to
2012 on a monthly time step to give a modern-day FW ($FW_{obs}$; annual max shown in Figure
1). Corresponding climate data on the same spatial and temporal resolution were obtained
from CRU-NCEP v4.0 (Wei et al. 2014) and averaged to give monthly values for a mean
modern-day year over the same time interval. The climate data for this mean year were then
used to drive two DGVMs: the Sheffield Dynamic Global Vegetation Model (SDGVM)
(Woodward et al., 1995; Beerling and Woodward, 2001*)* and the Lund-Postdam-Jenna model
(LPJ) (Wania et al., 2009) to produce corresponding vegetation data. The combination of
these yielded a reference data set of FW, climate (temperature and precipitation) and
vegetation (leaf area index, net primary productivity, transpiration, evapotranspiration, soil
water content and surface runoff) variables (either SDGVM or LPJ) for a set of $0.5° \times 0.5°$
spatial and monthly temporal resolution sites for a single modern-day average year. To ensure
that wetlands in areas dominated by agriculture or where one of our vegetation models,
SDGVM, predicts bare land, did not bias our FW predictions, such grid cells were removed
from the reference data. For the latter, this was done simply by removing those grid cells that
SDGVM predicted to be bare land. For the former, we removed those that were 50 % or
more, by cover, classed as cultivated and managed or mosaic cropland (Global Land Cover
2000 database, 2003).
Many of the methods that can be used to analyse the reference data and predict FW require
that the data are scaled, so that each variable covers a similar range of values. Therefore, we



scaled the values of each environmental variable, $X$, using their mean, $\mu_x$, and standard
deviation, $\sigma_x$, i.e. for a given grid cell, $J$, each variable was scaled as:
$$X'(J) = \frac{X(J) - \mu_x}{\sigma_x} \tag{1}$$
This scales all variables such that they have mean of 0 and standard deviation 1.

### 2.2 Test data sets

A modern-day test set was made by interpolating the reference climate data to $2.5° \times 3.75°$,
the spatial resolution often used for paleoclimate models. The DGVMs simulations were
conducted on this interpolated data to yield the vegetation outputs. All climate and vegetation
variables were scaled in the same way as the reference data, using the means and standard
deviations of the reference data. The palaeoclimatic assessment of our model was performed
using an early Eocene three dimensional fully dynamic coupled ocean-atmosphere global
climate model HadCM3BL-M2.2 (Valdes et al., 2017), on a 2.5° latitude by 3.75° longitude
grid and at a monthly time step for a single year. To simulate the early Eocene a Ypresian
paleogeography and high $CO_2$ (4x modern; 1120 ppm; Agnostous et al., 2016) was used.
SDGVM and LPJ were both run with these model-simulated climate data to produce the
vegetation variables required, as was done for the reference data set, whereas temperature and
precipitation were derived directly from the climate model. All variables were again scaled
using the means and standard deviations of the reference data. Therefore, for each climate,
modern day and early Eocene, we have two test data sets for a mean year on a monthly time
step, at 2.5° x 3.75° spatial resolution, both with the same climate data, one with SDGVM
vegetation data and one with LPJ vegetation data. Predictions for each test data set were
made with the corresponding vegetation model's reference data set.

### 2.3 Initial unsuccessful models of wetland fraction

Before discussing the model we employed to predict paleoclimate FW, it is useful to describe
briefly other strategies that we attempted but that did not yield robust predictions when
evaluated against modern-day data. The first of these was to examine FW vs individual
environmental variables graphically from the reference data, to ascertain if we could define
ranges for those variables that corresponded to predominantly low or high FW; this is similar
to the approach of Shindell et al. (2004), who proposed threshold values of standard deviation
of topography, ground temperature, ground wetness and downward shortwave flux for
wetland development. However, this proved unsuccessful, revealing only the rather obvious
relationship that wetlands do not usually occur when mean monthly temperature is below 0
°C. Although we expected to identify relationships for FW with other environmental
variables (i.e. ground wetness), none were found. This is due to the combined effects of
wetland occurrence being the function of multiple factors and the fact that most grid cells
have FW ≈ 0 for all months of the year and the number with significantly non-zero FW is
quite small. Therefore, environmental variables associated with high values of FW also tend
to be associated with FW ≈ 0. Poor correlation of FW with environmental variables is also
due to the important control exerted by the topography; regardless of climate, wetlands
cannot form in landscapes where excess water flows away rather than remaining in situ.



Collectively, these factors caused significant overlap in the range of environmental variables
associated with both low and high FW.
Another approach was a multiple linear regression using the reference data in order to derive
an equation for FW in terms of linear functions of multiple environmental variables.
However, this yielded equations that predicted a widespread occurrence of very low FW,
including those areas where $FW_{obs}$ is very high either seasonally or throughout the year.
Similarly, poor predictive models were obtained whether derived for all sites or just those
restricted to specific plant functional types. These outcomes likely occur because linear
regression optimises a function by minimising the error between predicted and observed
values. As most grid cells have FW ≈ 0 (Figure 1) the 'best' regression equation is one that
predicts FW very low almost everywhere, since in the majority of cases this is quite accurate.
Efforts were made to use other optimisation criteria with customised functions that attempted
to put more weight on predicting high FW correctly at the expense of larger errors where FW
is low. However, these simply over predicted FW. Therefore, we were unable to find any
satisfactory solution based on linear regression.

### 184   2.4 FW predicted by a nearest neighbour search

The reference data set of FW and environmental variables sites on a 0.5° grid at a monthly
time step can be viewed as a set of data points yielding FW at many different locations in a
multi-dimensional space. The eight dimensions of that space are the two climate and six
vegetation variables; temperature, precipitation, leaf area index, net primary productivity,
transpiration, evapotranspiration, soil water content and surface runoff. It is logical to assume
that points close to each other in such a space probably have similar FW. Therefore, if we
have the same environmental variables for a site of unknown FW, we can search the
reference data set for its nearest neighbour, i.e. the point nearest to it. We then predict it
would have the same FW as that for the nearest neighbour in the reference set, as illustrated
schematically below.
1. The set of N environmental variables, suitably scaled, $X_1, X_2 \ldots X_N$, defines an N-
196       dimensional space
2. The Euclidean distance between two points, $I$ and $J$, in this space is given by $D_{IJ}$

198       •  $D_{IJ} = \sqrt{\sum_{k=1,N}\big(X_k\,(I) - X_k(J)\big)^2}$         (2)

3. We calculate $D_{IJ}$ for site $I$ of unknown FW and all sites, $J$, in the reference data set,
200       for each of which we know FW($J$)
4. We find $J_{min}$, the nearest neighbour, that which gives the lowest $D_{IJ}$
5. We then predict FW ($I$) = FW ($J_{min}$)
6. If site $I$ is classed as bare land by the DGVM, thereby having all vegetation variables
204       = 0, we predict FW($I$) = 0

This nearest neighbour (NN) method can, if necessary, be extended to a KNN method,
whereby rather than predicting FW based solely on the single nearest neighbour we instead
consider some function of the K nearest neighbours.



## 2.5 Calculating wetland methane emissions

The aim of this study was to derive an algorithm for predicting wetland fraction that can then be used to calculate methane emissions. For the latter, we use the empirical method described by Cao et al. (1996), where methane production, *mp*, and methane oxidation, *mo*, rates for a specific grid cell and month are given by:

$$mp = R_h f_t \tag{3}$$

$$mo = mp \left( 0.6 + 0.3 \frac{GPP}{GPP_{max}} \right) \tag{4}$$

Where $R_h$ is soil respiration and *GPP* is gross primary productivity, both obtained from the respective vegetation model. $GPP_{max}$ is the maximum value of GPP for that grid cell for any month of the year. $f_t$ is a function that scales for temperature, *TMP,* in °C.

$$f_t = \frac{\exp(0.04055\,TMP)}{3.375} \tag{5}$$

This is capped at a maximum value of 1. In principle there would also be a scaling function for water table depth, but this is defined as 1 for inundated wetlands and we are only modelling inundated wetland fraction, as that is how the SWAMPS-GLWD FW dataset is defined.

Methane emission rate, *me*, is then the difference between methane produced and methane oxidised, scaled by the wetland fraction for that grid cell and month

$$me = (mp - mo)\,FW \tag{6}$$

## 3 Results and Discussion

### 3.1 Modern day test data set

The modern-day test set explained in Sect. 2.2 was used as a first, simple, test of the nearest neighbour algorithm for predicting FW described in Sect. 2.4. Since the modern-day test set is simply the reference climate data downscaled from 0.5° to the courser HadCM3BL-M2.2 model grid of 2.5° by 3.75° (with vegetation from the DGVMs), we expect the NN algorithm to yield predicted FW reasonably consistent with a similar downscaling of the SWAMPS-GLWD observed FW. If the NN predicted FW does not achieve this, then that would indicate that the NN algorithm has failed to predict FW sufficiently accurately. Therefore this test is primarily designed to indicate that a nearest neighbour algorithm either does or does not have the potential to be applied to paleoclimates.

Fig. 2 shows maps of seasonal, June–July–August and December–January–February, average FW from the observed SWAMPS-GLWD data interpolated to 2.5° x 3.75° along with the predicted FW using either SDGVM or LPJ vegetation data test sets. For both vegetation models, the predicted FW maps are similar to the observed-interpolated data. Sparse patches of high FW occur in the tropics, especially the Amazon, throughout the year, and large areas of seasonal summer wetlands occur in Alaska, Canada and Siberia. The monthly variation of FW north and south of 30° N, i.e. essentially comparing boreal and tropical wetlands is shown in Figure 3. We split the global values into these two zones because there are virtually





no southern hemisphere boreal wetlands, and any division based purely on latitude is
arbitrary. The nearest–neighbour algorithm generates the correct seasonal FW pattern in
boreal regions and, as expected, a relatively constant monthly FW in the tropics. However,
SDGVM consistently underestimates the amount of tropical wetland, whilst LPJ agrees
reasonably well with observations; mean monthly values are 2.11, 1.47 and 1.90 x $10^6$ km$^2$
for the observed, SDGVM and LPJ respectively. This is due to the fact that SDGVM classes
some grid cells as bare land, assumed to have FW = 0 in our algorithm, even though some of
these have non-zero FW in the SWAMPS-GLWD database. LPJ does not classify these grid
cells as bare land but instead treats them as very low amounts of vegetation, therefore
yielding higher global FW that is more consistent with observations. If we exclude from the
observed data those grid cells SDGVM predicts as bare land, then the SDGVM prediction
matches better the observed data and LPJ predictions (Table 1). These results give confidence
that a nearest neighbour algorithm is able to reproduce acceptable FW based on these specific
climate and vegetation variables.
Figure 4 shows the monthly variation in wetland methane emissions for boreal and tropical
areas, calculated using: the observed or predicted FW, both vegetation models' outputs and
Eq. 3 to 6. The annual methane emissions totals are summarised in Table 2, along with other
recent estimates from model intercomparisons. The annual and monthly zonal methane
emissions are broadly similar for a given vegetation model regardless of whether the
observed or predicted FW is used. SDGVM gives global emissions in line with the other
modelling studies, whereas those from LPJ are somewhat lower. This is mainly due to
differences in tropical emissions. SDGVM yields higher tropical emissions than LPJ but
slightly lower emissions north of 30°N. The main factors influencing the modelled methane
emissions (other than FW) are, according to equations (3) to (5), temperature (which is the
same for both vegetation models), soil respiration ($R_h$) and gross primary productivity ($GPP$),
the latter two differing between the two vegetation models. It appears that differences in $R_h$
lead to the different zonal methane totals. South of 30° N SDGVM and LPJ model annual
total $R_h$ of 46,000 Tg C year$^{-1}$ and 35,000 Tg C year$^{-1}$ respectively and, using the same
observed FW, SDGVM and LPJ model annual methane emissions of 123 Tg CH$_4$ year$^{-1}$ and
69 Tg CH$_4$ year$^{-1}$ respectively. Therefore, in the tropics the differences in the predicted
methane emissions seem to be due to differences in calculated $R_h$. North of 30° N both
DGVMs have similar $R_h$, 20,000 Tg C year$^{-1}$ and 22,000 Tg C year$^{-1}$ respectively for
SDGVM and LPJ, and similar values of methane emissions, 64 Tg CH$_4$ year$^{-1}$ and 65 Tg CH$_4$
year$^{-1}$ respectively.

**3.2 Early Eocene climate**

In the previous section we have shown that a NN method can reproduce FW for a modern
day climate, justifying its application to the early Eocene climate described in section 2.2.
However, as noted at the end of section 2.4 a NN method can be extended to KNN, whereby
we predict FW based on some function of the FW of K nearest neighbours (noting that in 3.1,
NN is simply 1NN, i.e. KNN with K=1). A 1NN algorithm that works well to predict modern
day FW may not work as well for a paleo climate of many millions of years in the past. The
reference data set we use, section 2.1, is very similar to the modern day test set, the latter's
climate data is simply obtained by interpolating the former to a courser spatial grid.
Therefore, we expected and observed high correlation between modern day FW predicted
from the nearest neighbour in the reference data and the actual FW.  The early Eocene test



data has significant differences to the reference data since the climate of the early Eocene is
obviously not the same as the modern day. Therefore, it will be harder for a nearest neighbour
based method, searching a space described by climate and vegetation data, to find a nearest
neighbour in the modern day reference data with the correct early Eocene FW, whatever that
may be. It may be that for a high FW early Eocene grid cell the nearest neighbour happens to
have quite low FW and vice versa. Figure.1 shows that FW can change from very high to
almost zero over relatively small distances, for example in the Amazon basin, and that
therefore sites with similar climate and vegetation can have very different FW. The greater
the degree of difference between the early Eocene and the modern day reference data sets, the
more likely it is that the first nearest neighbour does not have the correct FW.
FW calculated for the Early Eocene using the exact same 1NN method as used for the
modern day test set yields values of global monthly mean wetland area of 4.07 x $10^6$ km$^2$
using SDGVM. This is around 33% higher than that for the modern day, 3.00 x $10^6$ km$^2$ from
Table 1.  However, this includes a contribution of 1.53 x$10^6$ km$^2$ from areas south of 30° S,
which have an almost negligible contribution for the modern day, so the tropics and northern
Boreal regions actually have lower FW for the Early Eocene. Given that the Early Eocene
was significantly warmer and wetter than the modern day (Carmichael et. al. 2017), we
expect greater wetland area than the modern day. Beerling et al. (2011) reported global
wetland area for an Early Eocene climate using SDGVM; employing their method to our
Early Eocene climate, so as to eliminate differences arising from the specific HadCM3 model
climate and spatial resolution, yields global monthly mean FW area of 16.29 x $10^6$ km$^2$, four
times higher  than the value we would calculate from a 1NN method. Therefore, based on
comparison with both the modern day and a previous Eocene study, it appears that a 1NN
method may be unsuitable for a paleoclimate that is very different to our modern day
reference climate, and we consider KNN with higher values of K.

### 318   3.2.1 maxKNN FW prediction

If indeed the 1NN results are too low then that implies that for some hypothetical high FW
sites from the Early Eocene, the first nearest neighbours in the reference data have very low
FW. Therefore, if we consider higher values of K we may improve our estimate by predicting
FW to be the maximum FW of K nearest neighbours in the reference data. However,
applying this approach will yield  increasingly higher FW as K increases,  requiring a data-
constrained optimisation of K. Here we use the distribution of coal deposits in the Eocene,
(Boucot et al., 2013) shown in Figure 5 as such constraints. There are some limitations to this
approach. Coal is formed in wetlands, but can also form in other settings such as lakes; and of
course, these datasets do not document where wetlands were present but the sedimentary
record is missing or has not been published. In the tropics, coal may not have formed in
wetland environments due to a very high rate of carbon cycling and in northern latitudes
subsequent glaciations could have eroded coal deposits away. Moreover, data will be sparse
or non-existent for remote or inaccessible modern day regions, such as under the Antarctic
ice sheet. We also note that precise age and location, especially when comparing to low
resolution climate simulations, could cause disagreement for grid-by-grid comparisons. A
final and critical complication is that FW is a number between 0 and 1, corresponding to the
fraction of a site that is wetland, whereas the coal data is a binary measure: either a grid cell





has or does not have a coal deposit within it. For all of these reasons, data-model comparisons
must be done cautiously; nonetheless, these data are useful for identifying the most effective
K value for reconstructing likely wetlands.
We defined two functions to assess how well a model FW matched the locations of Eocene
coal deposits. Firstly, *f1* is defined as the mean distance, in km, of a coal deposit location to a
grid cell  with model FW predicted to be > 0.2. The choice of 0.2 representing significant FW
is arbitrary but the analysis was repeated with other values and the same conclusions were
found. Secondly, *f2* is defined as the mean FW of the grid cell closest to each coal deposit
location, providing that site is within 2 grid points of that coal deposit location, to allow some
leeway with regard to different projected locations of land masses in the early Eocene. Again
the choice of a 2-pixel limit is arbitrary but the analysis was repeated with other limits and
the same conclusions found.
Figure 6 shows the values of *f1* and *f2* for maxKNN predictions of FW with increasing K for
both the SDGVM and LPJ Early Eocene data sets, compared to a data set of coal deposit
locations. As explained, since FW increases with K then by extension, so does the likelihood
of a site with a coal deposit in or close to it coinciding with a site of significant FW.
Therefore, we do not seek to find the value of K that will give the lowest value of *f1* and
highest value of *f2* as that would simply be K equal to the size of the entire reference data set.
Instead, we try to find the lowest value of K that gives a "good" prediction for both *f1* and *f2*.
Although "good" is a subjective measure, we define it based on where increases in K result in
marginal improvements in f1 and f2. For both vegetation models as K increases from 1 to 3 *f1*
decreases significantly and *f2* increases significantly. For K > 3 the decrease in *f1* levels out
and the increase in *f2* also declines. Therefore, we conclude that based on comparison of
predicted FW and locations of coal deposits, K=3 is a reasonable choice to make predictions
for our early Eocene climate via a maxKNN algorithm.

**3.2.2 FW predicted by max3NN**
Figure 7 shows annual maximum FW (i.e. for each pixel the highest of the 12 monthly
values) calculated by a max3NN model using SDGVM or LPJ vegetation data, as described
above, with the locations of early Eocene coal deposits also shown. The annual maximum
FW is shown here as FW might only need to be high at some point during the year to give
rise to coal deposits.  The areas of predicted high FW are much larger than for the modern
day (Fig. 1); moreover, at this spatial resolution there are often abrupt changes from low-
medium (yellow) to much higher (red) values leading to some isolated patches of high FW.
The approach makes it difficult to interrogate specific factors that drive the increase in
Eocene FW compared to today but given the wetter climate of the Early Eocene higher FW
than the modern day is to be expected. The patchiness is partly a consequence of using annual
maximum FW but also reflects the challenge of predicting a characteristic of a
paleoenvironment based on modern day reference data. Considering zonal total FW and
seasonal average FW maps, i.e. averaging out some of the small scale spatial and temporal
variability, is likely a better approach for understanding ancient methane cycling and these
are discussed later.





The maps of predicted FW are quite different for the two vegetation models, but the greatest
differences are in areas with very little or no coal deposits, e.g. the tropics, north eastern
North America and Antarctica, making it difficult to critically evaluate them against the data.
However, the monthly variations given by the two vegetation models in total FW (Figure 8)
and methane emissions (Figure 9), for the three latitudinal zones are reasonably similar with
respect to seasonal variations, in that both have their highest values in the summer months for
zones north of 30° N and south of 30° S and no clear seasonal variation in the tropics. In the
tropical zone, predictions of monthly FW area are similar in magnitude for the two vegetation
models, with SDGVM usually predicting higher FW than LPJ. However, in the zone north of
30° N LPJ predicts much higher FW than SDGVM throughout June to October with a peak in
September, whereas SDGVM peaks in May. A similar but less striking pattern occurs for the
zone south of 30°S where again LPJ predicts higher summer FW area than SDGVM. These
differences between the two vegetation models are also evident in maps of seasonal average
predicted FW (Figure 10). In June to August, SDGVM predicts very little wetland area in the
northern hemisphere, whereas LPJ predicts moderate to high FW areas over much of the land
north of around 50° N. In December to February both models predict almost zero FW north
of around 50° N. In the tropics and the southern hemisphere, the two models predict similar
amounts of wetland area, but with SDGVM predicting slightly higher FW overall between
30° S to 30° N and LPJ predicting slightly higher FW south of 30° N.
This differs from the modern day distribution of wetlands (Figure 1) and likely arises from a
variety of method-dependent factors. First, the coarser resolution leads to more patchy
distribution, as is evident in the modern day data in Figures 1 and 2 (top row) at 0.5° x 0.5°
and 2.5° x 3.75° spatial resolution. This is particularly true for the tropics where wetlands do
occur in small areas. Secondly, the nature of the nearest neighbour algorithm relies on the
principle that a grid cell in a paleoclimate with specific values of environmental variables will
have the same FW as a grid cell in a modern day reference data set with similar values for
those environmental variables; however, other factors influence wetland fraction, such as the
topography. Therefore, a nearest neighbour method predicting FW for a paleoclimate from a
modern day reference data may well have errors for a given grid cell and month. These errors
should reduce when averaged over latitudinal zones or seasonal averages.
The differences between methane emissions from the two vegetation models likely arise from
their respective impacts of soil water balance, via the magnitude of evapotranspiration (EVT)
relative to precipitation (PRC). As the vegetation and climate models are not dynamically
coupled, PRC will be the same in all Eocene simulations, but EVT will vary; thus, vegetation
models that yield elevated EVT in a given grid cell are more likely to yield negative water
balance (PRC-EVT) and low FW. Figure 11 shows the June to August mean PRC-EVT for
SDGVM and LPJ, revealing that it is negative in most places north of 30° N for SDGVM but
is slightly positive or at least much closer to zero for LPJ. Therefore, SDVGM will generally
predict lower FW by identifying modern day nearest neighbours where PRC < EVT and
unlikely to be wetland. The lack of extensive coal deposits in the high northern latitudes,
especially where the LPJ-based approach predicts wetlands, could indicate that the LPJ
approach has over-predicted FW. However, we caution that this could be a data limitation
issue and future work is required to interrogate the forecasts of these two methods.
Regardless, both models yield broadly similar results on global and zonal terms (Table 3)
indicating that the KNN algorithm could be a useful complementary approach for





interrogating ancient wetland extent and methane emissions. Global monthly mean FW is 8.5
x $10^6$ km$^2$ and 10.3 x $10^6$ km$^2$ predicted by SDGVM and LPJ respectively. Both of these
values are larger than for the modern day value of 3.0 x $10^6$ km$^2$, as we would have expected.

### 4. Conclusions

We have presented a nearest neighbour method by which FW can be calculated at sites on the
Earth's surface for an Eocene paleoclimate based on a set of environmental variables
obtained from climate and vegetation models and comparison of these to a modern day
reference data set. The precise formulation of the nearest neighbour approach was determined
through comparison to locations of Eocene coal deposits and indicated that a max3NN
method was best suited in this case. That should not be taken to imply that a max3NN would
be the best in general; for another paleoclimate a similar analysis to that performed here
would be required to determine the optimum implementation of KNN. The predicted
distributions of FW are much higher than those of today, as we would expect. We have
assessed this using two different global vegetation models, and whilst these do yield some
geographical differences in FW arising from different evapotranspiration estimates, they are
broadly similar when considering zonal means. For both vegetation models, global monthly
mean modelled FW area is less than, around half to two thirds, that of Beerling et al., 2011,
as are the values of the wetland methane emissions. However, our new method does not rely
on the standard deviation of orography, a variable which is only known to a relatively coarse
resolution for deep paleoclimates.

### Code and Data

This study presents a methodology using existing data and climate and vegetation models.
Information relating to these is already included in this article. Code implementing the
maxKNN prediction of FW is included as supplement.

### Author Contribution

DJW and DJB planned the work with advice from all co-authors. DJW carried out most of
the experimental work with MB providing the HadCM3BL-M2.2 and EPK the LPJ model
data. DJW prepared the manuscript with contributions from all co-authors.

### Competing Interests

The authors declare that they have no conflict of interest.

### Acknowledgements

Funding was provided by the Natural Environmental Research Council (NERC) grant
NE/J00748X/1. The authors would like to thank Chris Scotese for access to and advice on
Eocene coal deposit data.



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



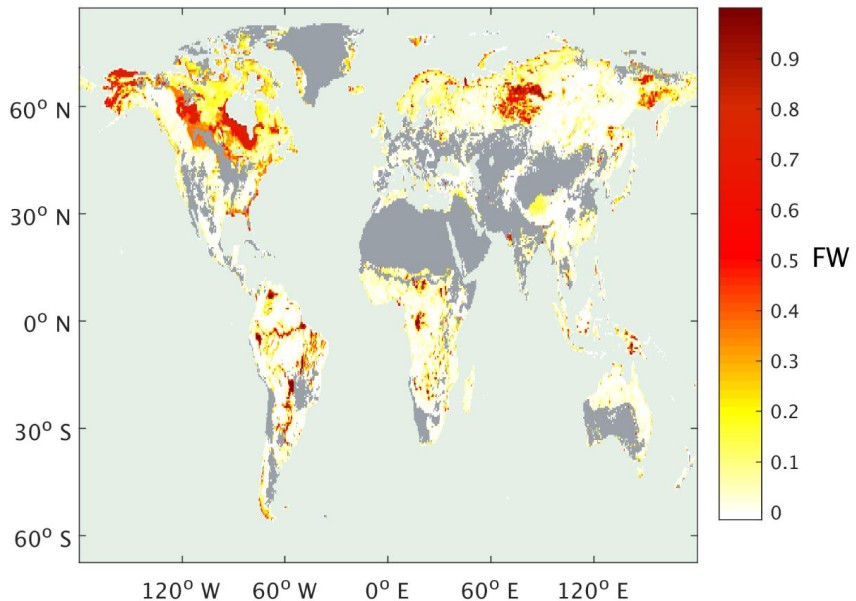

**Figure 1**: **Annual maximum observed FW from the SWAMPS-GLWD data set (Poulter et. al., 2017), mean of 2000 to 2012. Grey shading indicates bare land, as predicted by SDGVM, or > 50% cultivated (Global Land Cover 2000 database, 2003).**





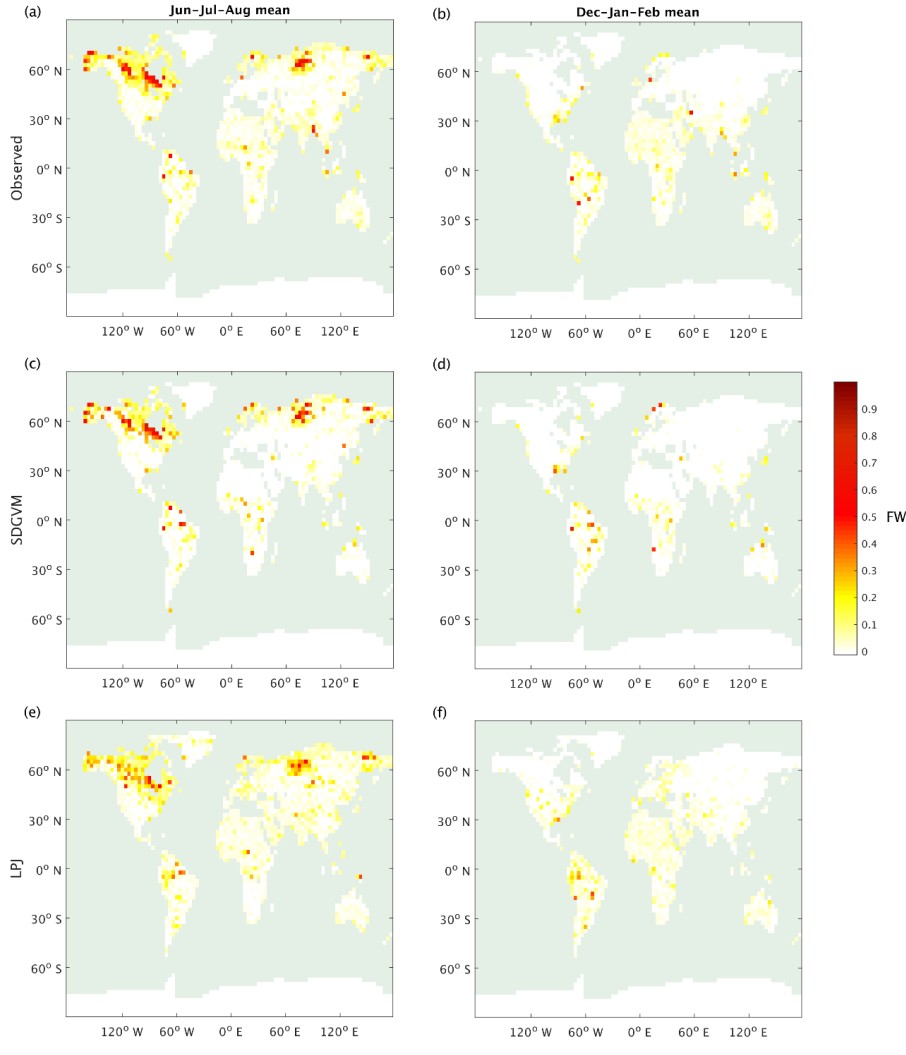

**Figure 2: Seasonal mean FW. Observed interpolated to model grid; (a) Jun–Jul–Aug
and (b) Dec–Jan–Feb. 1NN prediction by SDGVM (c) Jun–Jul–Aug and (d) Dec–Jan–
Feb. 1NN prediction by LPJ (e) Jun–Jul–Aug and (f) Dec–Jan–Feb.**


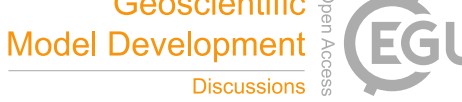

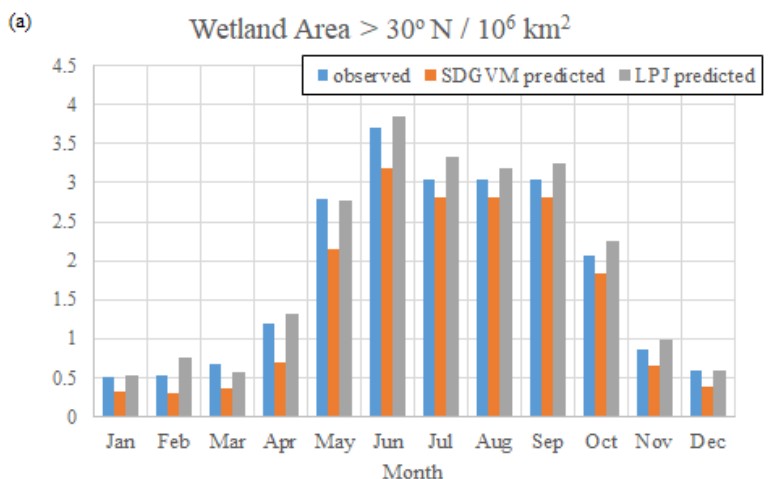

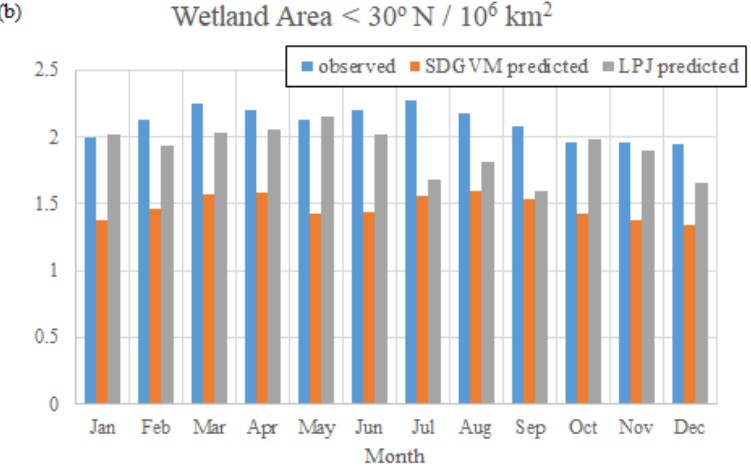

**Figure 3: Monthly zonal variations of FW calculated for the mean 2000-12 climate on a**
**2.5 x 3.75° grid, (a) North of 30° N and (b) South of 30° N.**





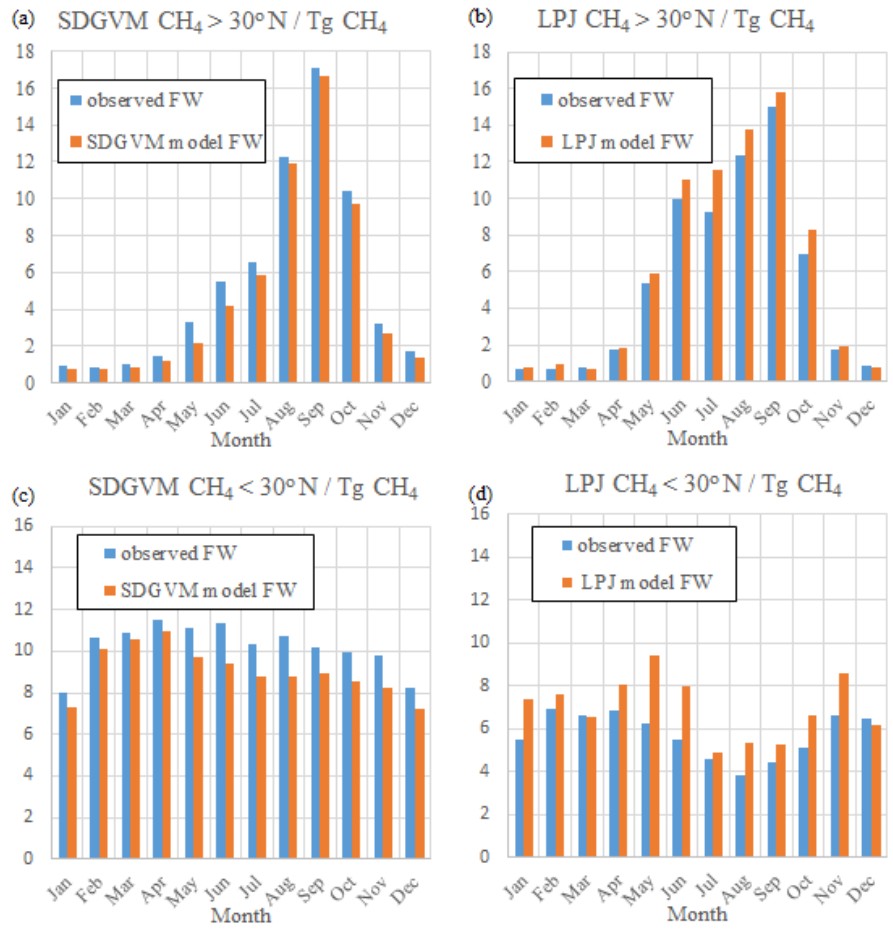

**Figure 4: Monthly zonal variations of wetland CH4 calculated from DGVM model data**
**and observed or modelled FW, for the mean 2000-12 climate on a 2.5 x 3.75 ° grid. (a)**
**SDGVM North of 30° N, (b) LPJ north of 30° N, (c) SDGVM South of 30° N and (d)**
**LPJ south of 30° N.**





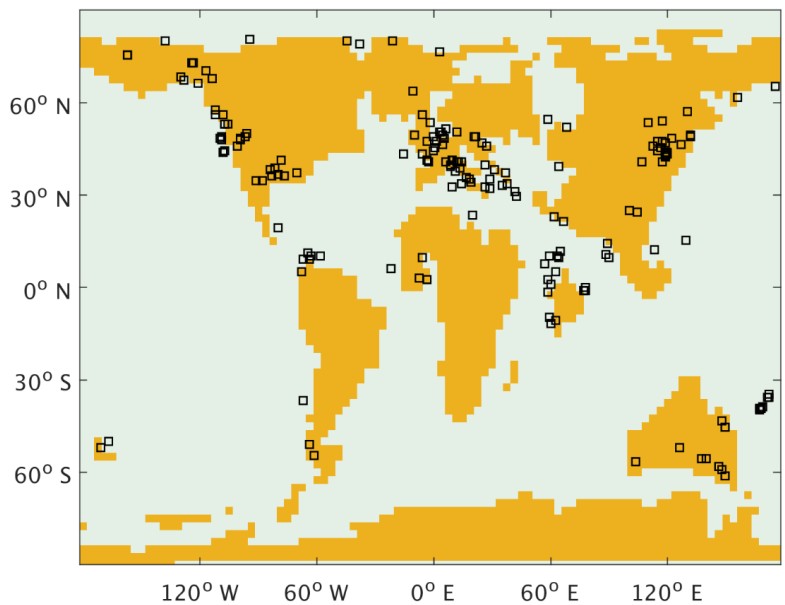

**Figure 5: Locations of Eocene coal deposits plotted on our Eocene model land mask.□**
**indicates an Eocene coal deposit location (Boucot et al., 2013)**

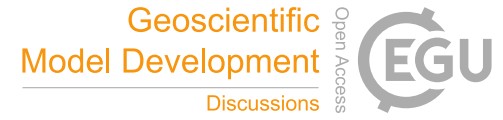



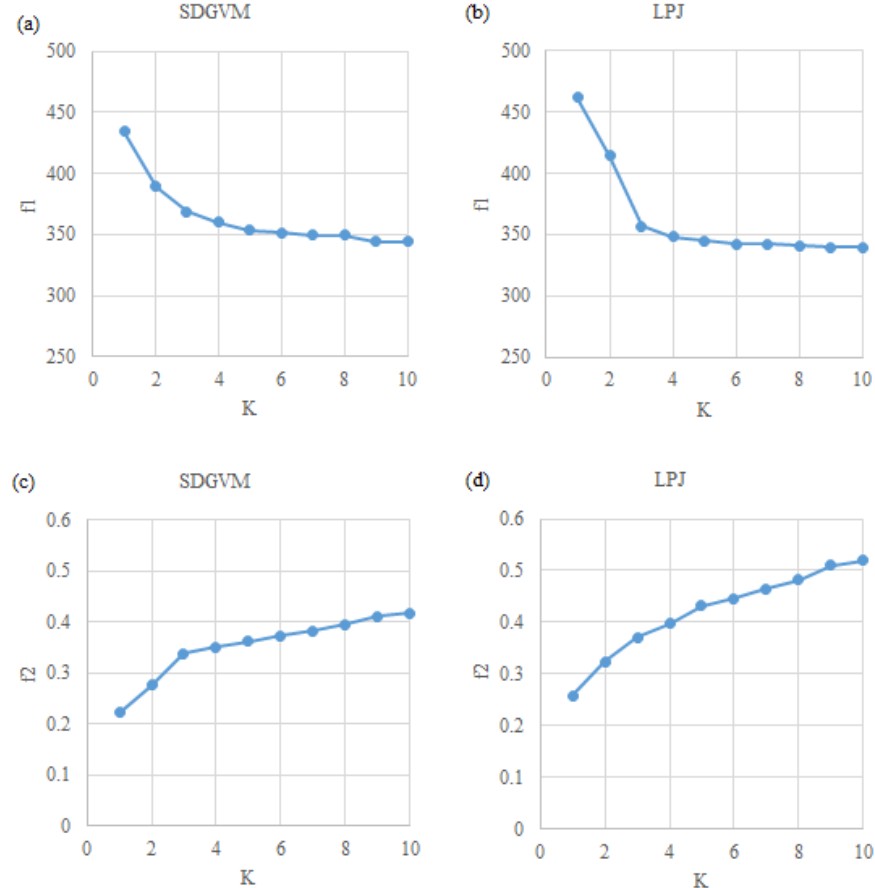

**Figure 6: Variations of statistics for match between Eocene maxKNN predicted high**
**FW and coal locations (Boucot et al., 2013). f1 is the mean distance of a coal location to**
**site with FW > 0.2 for model based on (a) SDGVM and (b) LPJ. f2 is the mean FW of**
**sites within 2 pixels of a coal location for model based on (c) SDGVM and (d) LPJ data.**





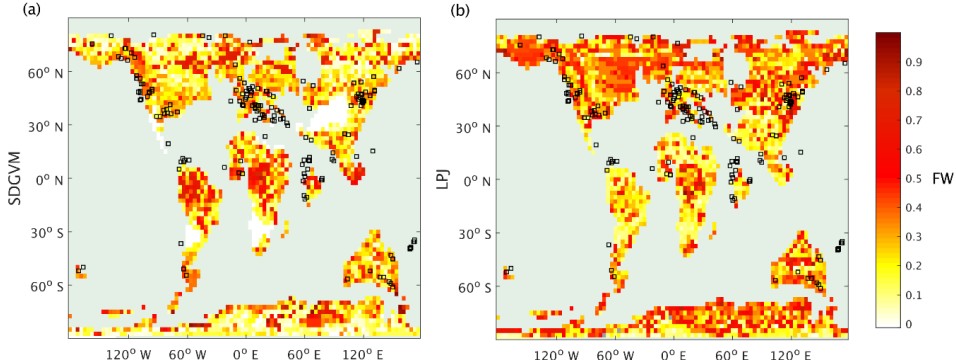


**Figure 7: Annual maximum FW calculated by the max3NN method by SDGVM and LPJ for the Eocene climate, compared with coal deposit locations**




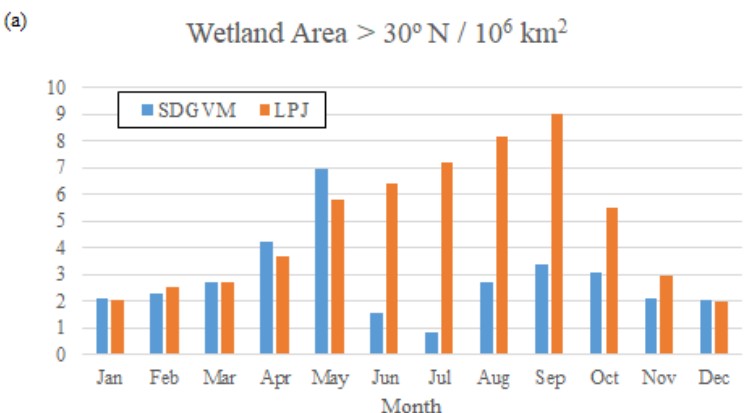

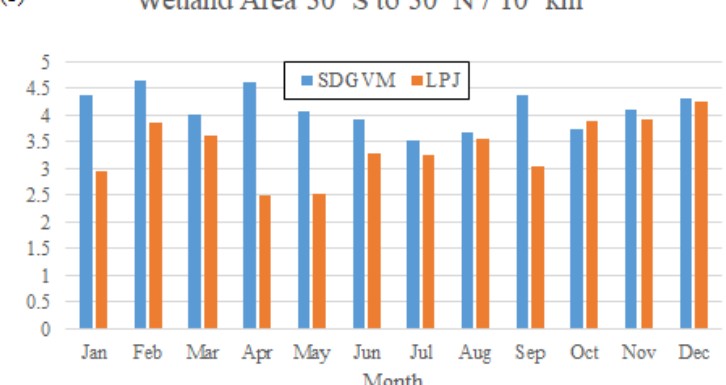

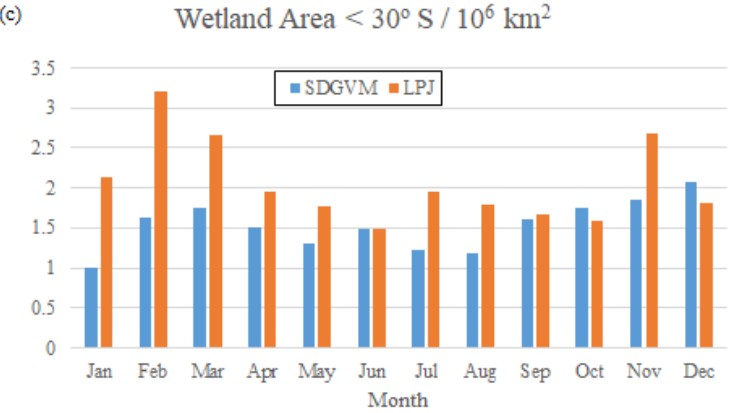

**Figure 8: Monthly variations of total wetland area calculated for the Eocene climate by SDGVM and LPJ, for (a) all areas north of 30° N, (b) all areas between 30° S and 30° N and (c) all areas south of 30° S.**

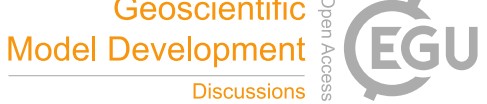

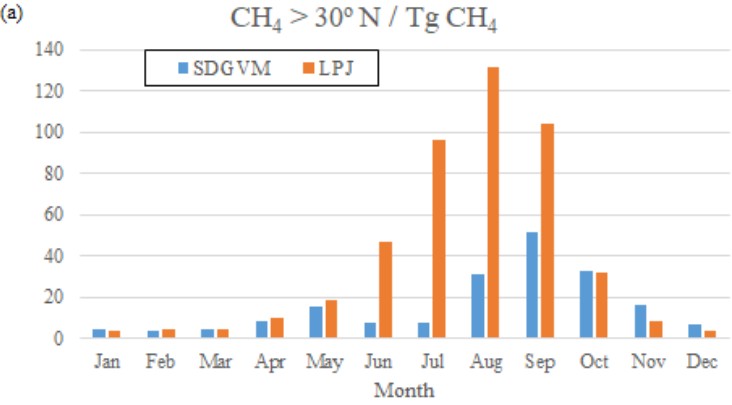

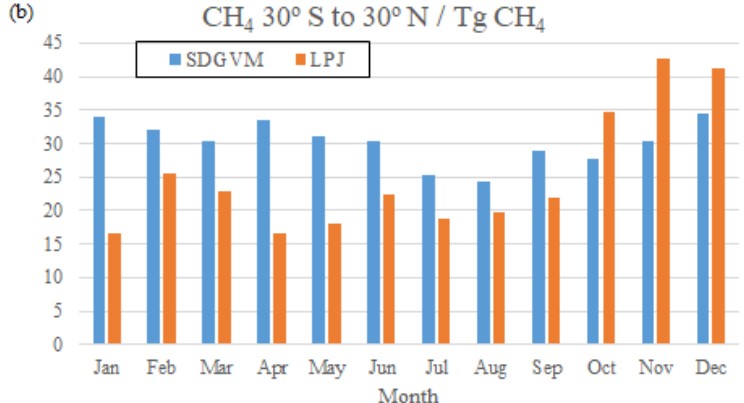

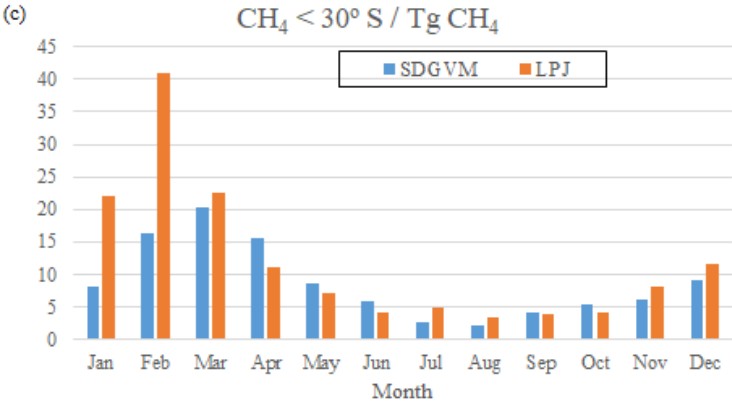

**Figure 9: Monthly variations of wetland CH₄ calculated from predicted FW, for the**
**Eocene climate by SDGVM and LPJ, for (a) all areas north of 30° N, (b) all areas**
**between 30° S and 30° N and (c) all areas south of 30° S.**



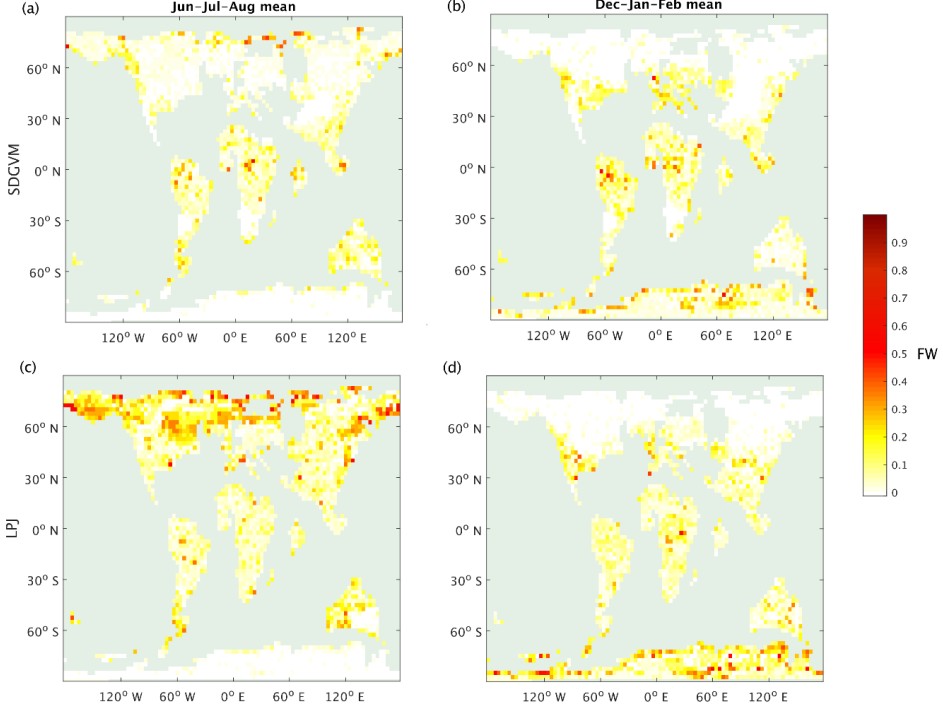


**Figure 10: Seasonal mean FW predicted for the Eocene climate by SDGVM and LPJ using the max3NN (a) SDGVM June–July–August, (b) SDGVM December–January– February, (c) LPJ June–July–August, (d) LPJ December–January–February**






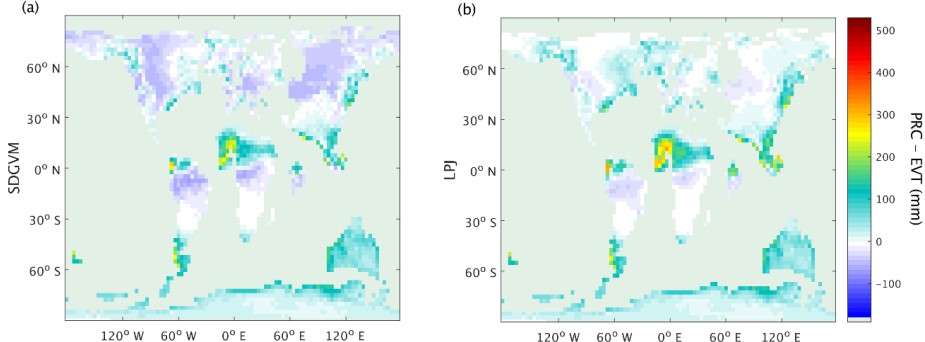

**Figure 11: June–July–August mean precipitation minus evapotranspiration for the Eocene climate, using evapotranspiration from (a) SDGVM or (b) LPJ.**





| | > 30° N FW | < 30° N FW | Global FW |
|---|---|---|---|
| **Observed** | 1.84 | 2.11 | 3.95 |
| **Observed excluding SDGVM bare land** | 1.47 | 1.41 | 2.88 |
| **SDGVM** | 1.53 | 1.47 | 3.00 |
| **LPJ** | 1.95 | 1.90 | 3.86 |


**Table 1: Modern day monthly mean FW area ($10^6$ km$^2$), for observed data interpolated**
**to the 2.5° x 3.75° grid or calculated by vegetation model.**



| Model | FW data | > 30° N CH$_4$ | < 30° N CH$_4$ | Global CH$_4$ |
|---|---|---|---|---|
| **SDGVM** | **observed** | 64.32 | 122.69 | 187.01 |
| | **predicted** | 57.95 | 108.63 | 166.58 |
| **LPJ** | **observed** | 65.43 | 68.60 | 134.03 |
| | **predicted** | 73.11 | 83.78 | 156.89 |
| | | | | |
| **GCP-CH4**[*] | **observed 0.5°** | | | ~ 184 |
| **WETCHIMP**[**] | **model specific** | 51±15 | 126±31 | 190±39 |


[*] GCP-CH4 (Poulter et al., 2017) results are the mean of 11 different methane emission
models with the same observed wetland data as used to produce Figure 1 here. They are
quoted as means over specific ranges of years; 2000–2006 = 184.0 ± 21.1, 2007–2012 =
183.5 ± 23.1, 2012 = 185.7 ± 23.2. As our results are for a single mean 2000–12 year we
therefore only quote an approximate value from this source for comparison.
[**] WETCHIMP (Melton et al., 2013) results are the mean of 8 different models, 1993-2004,
each of which used their own definition of wetland extent rather than observed data

**Table 2: Modern day annual total wetland CH$_4$ emission (Tg CH$_4$ year$^{-1}$), calculated by**
**vegetation model using either observed FW data (interpolated to the 2.5° x 3.75° grid)**
**or model predicted FW, compared with other modelling studies.**






| FW model | > 30°N | 30°S to 30°N | < 30°S | Global |
|----------|--------|--------------|--------|--------|
| **SDGVM** | 2.82 | 4.11 | 1.53 | 8.48 |
| **LPJ** | 4.84 | 3.39 | 2.06 | 10.29 |

**Table 3: Eocene monthly mean max3NN modelled FW area / $10^6$ km$^2$**