# Peer review of "A Predictive Algorithm For Wetlands In Deep Time Paleoclimate Models"

_Geoscientific Model Development, 2018_

## Referee Comment (RC1) · Anonymous Referee #1 · 1 Dec 2018

The manuscript by Wilton et al. covers an interesting topic appropriate for GMD. The authors attempt to develop a nearest neighbor-based algorithm to simulate the inundation dynamics for estimating wetland CH4 emissions in deep time paleoclimate. The writing is clear. The results are interesting and this approach provides a way to simulate wetland areal dynamics in ancient climate. However, there are several issues in this manuscript needed to be addressed before publication.

The main confusion I have is on the validation of this approach. It is not convincing that using one reference dataset to train their algorithm, and then evaluate the simulated results with the same reference dataset. It would be necessary to compare with independent inundation products to justify their approach, or the authors need to provide the uncertainty in the estimated inundation using their approach given that there are

large uncertainties in wetland extent among existing inundation products (Melton et al., 2013).

-The logic of this approach is a bit confusing to me. If I understand it correctly, this nearest neighbor-based algorithm implicitly assumes the locations of wetlands should close to each other and inundation is correlated with eight variables the authors proposed. But according to the modern dataset, is there any analysis/evidence prove that this relationship exist? Fan (2011) suggest that water table depth is a key to simulate wetland distribution - at least it is an important variable to capture the distribution of peatlands in high latitudes as some of the peatlands don't show inundated condition but still emit CH4.

- I'm not sure that comparing the simulated wetland distribution with coal deposit can be helpful as the authors have already mentioned some of the limitations using coal deposit. Also, it's hard to tell how good the fit is from reading Figure 7.

- It would be great to address a bit more about the background why it's important to develop a dynamic inundation algorithm for deep time paleoclimate simulation and what's the current status of research on this topic.

References:

Fan, Y., and G. Miguez-Macho. 2011. A simple hydrologic framework for simulating wetlands in climate and earth system models. Climate Dynamics 37:253-278.

Melton, J. R., R. Wania, E. L. Hodson, B. Poulter, B. Ringeval, R. Spahni, T. Bohn, C. A. Avis, D. J. Beerling, G. Chen, A. V. Eliseev, S. N. Denisov, P. O. Hopcroft, D. P. Lettenmaier, W. J. Riley, J. S. Singarayer, Z. M. Subin, H. Tian, S. Zürcher, V. Brovkin, P. M. van Bodegom, T. Kleinen, Z. C. Yu, and J. O. Kaplan. 2013. Present state of global wetland extent and wetland methane modelling: conclusions from a model inter-comparison project (WETCHIMP). Biogeosciences 10:753-788.

---

## Author Comment (AC1) · 14 Dec 2018

We thank the referee for their comments and respond to the points they raise below.

1. "The main confusion I have is on the validation of this approach. It is not convincing that using one reference dataset to train their algorithm, and then evaluate the simulated results with the same reference dataset. It would be necessary to compare with independent inundation products to justify their approach, or the authors need to provide the uncertainty in the estimated inundation using their approach given that there are large uncertainties in wetland extent among existing inundation products (Melton et al., 2013)." There is no training and evaluation in the sense that would normally be understood from a machine learning perspective. For the Eocene results, section 3.2,

we clearly have no wetland data with which to train and evaluate our predictions. We simply use the coal deposits as a proxy, comparing those to our wetland predictions to give us the best value of K for the maxKNN approach with this particular data set. We are happy to improve the text in this section to make this clearer.

Nor are we using a training set for the modern day test data, section 3.1. These results were included simply to show whether some form of nearest-neighour approach might, in principle, be useful (lines 236-238); we were exploring the potential of this approach. It was a test that if failed would have meant we would not have continued developing a nearest neighbour method; it would have been another unsuccessful attempt along with those briefly discussed in section 2.3. That the method passed this test merely indicated we could explore some form of nearest neighbour method in the context of the Eocene climate. If this is what the referee is referring to, then we will improve the text in section 3.1 to make this clearer.

2. "The logic of this approach is a bit confusing to me. If I understand it correctly, this nearest neighbor-based algorithm implicitly assumes the locations of wetlands should close to each other and inundation is correlated with eight variables the authors proposed. But according to the modern dataset, is there any analysis/evidence prove that this relationship exist" The nearest neighbour approach assumes that sites with similar values of wetland fraction should have some similarity in terms of their values of the 8 climate & vegetation variables we use; or to put it another way, if sites with similar FW show no similarity at all between their values of at least some of those 8 variables, then a nearest neighbour approach will simply not work. There is certainly no simple correlation between FW and those 8 variables in the modern day data, as we briefly explain in our "Initial unsuccessful models" section 2.3; a multiple-linear regression on those 8 variables did not produce a good predictive model of FW. This suggests that any relationship between FW and those 8 variables must be complex. We are happy to add to the text to further explain this, the best place probably being at the end of section 2.3.

3. "Fan (2011) suggest that water table depth is a key to simulate wetland distribution - at least it is an important variable to capture the distribution of peatlands in high latitudes as some of the peatlands don't show inundated condition but still emit CH4."

We use soil water content, defined as the amount of water in in the top 1m of soil. This is produced by both vegetation models whereas water table depth is not.

4. "I'm not sure that comparing the simulated wetland distribution with coal deposit can be helpful as the authors have already mentioned some of the limitations using coal deposit. Also, it's hard to tell how good the fit is from reading Figure 7." Clearly coal deposits are not an ideal proxy for wetland fraction, but they are all we have. Without them we would have had no way of deciding on a value for K in the maxKNN algorithm. Therefore, despite the limitations, they are useful to explore this approach.

5. "It would be great to address a bit more about the background why it's important to develop a dynamic inundation algorithm for deep time paleoclimate simulation and what's the current status of research on this topic." As explained in the introduction, there is great interest in understanding how the extent of wetlands changed through geological time and what role that could have had on methane cycling. However, there is currently only one model-based approach for deep time paleoclimates (Beerling et al., 2011). The goal of this paper is to explore other methodologies and compare them to this original work, better understanding the potential of the new approaches and the robustness of the previous work. We will add additional text to the introduction to address this further.

---

## Referee Comment (RC2) · Anonymous Referee #2 · 21 Dec 2018

Review of 'A predictive algorithm for wetlands in deep time paleoclimate models', by David J. Wilton et al.

**General comments**

This paper attempts to produce a statistical model ("predictive algorithm") to estimate the global distribution of paleo-wetlands, and the associated methane emissions. The method (as far as I understand it) is to look at present-day wetlands, and find relationships between various climatic and vegetation variables that can be used to predict wetland extent. It turns out to be necessary to not just look at driving environmental variables at the same location, but to look at nearest neighbours (in parameter space – not spatially). A nearest neighbour model (NN) using just one NN is found to well represent present-day wetlands. However, the authors find that this approach applied

to Eocene climate model data doesn't produce a very realistic wetland distribution – based on proxy evidence from coal deposits. Using multiple nearest neighbours improves the model performance – three appears optimal.

I am fairly convinced this is a sensible and useful approach, but I must admit to being slightly baffled about the exact methods employed – I found the paper rather unclear in quite a few places. I would encourage the authors to revise the description of the methods to make it clearer. Some clarification on how this approach should be employed by the wider modelling community would also be appreciated – can the method be embedded within ESMs to calculate wetland emissions online? Or is it envisaged as an offline only tool? I wasn't clear.

If the authors can clarify the methods and address the specific points below, then I believe this will be a very useful method for modellers to use.

Specific comments

L43 ESMs must either prescribe CH4 concentrations as boundary conditions, or "incorporate dynamic methane fluxes from natural sources...". If the latter, they must not only simulate the sources but also the sinks of the CH4 (i.e the whole budget) in order to reasonably represent concentrations.

L55 '...no direct observations of wetland extent' – it should be stated that there are however proxies, that you later utilise (i.e. coal deposits).

L60 '...mean monthly temperature drops below 0 °C at some point in the year...' I found this slightly confusing. Do you mean if there is one (or more) month in the year below 0 °C, then that grid-cell is classified as producing methane? Clarify.

L71 So you are using DGVMs to simulate vegetation distributions, rather than using present-day observational datasets. It may be worth saying that the DGVMs have (presumably) been evaluated elsewhere.

L68 Is it worth briefly defining wetland? Perhaps earlier. E.g. the RAMSAR definition.

GMDD
Is it obvious how such definitions translate into a climate model-specific definition? (Water depth, etc.). What is the basis of the modern day reference data set of FW? Can you say it is 'known' or 'observed' FW?

L80 Typo: intercomparson

L87/90 Capitalise Nearest Neighbours or not? (Is it a well enough known method to be considered a proper noun? I don't know, but at least be consistent.)

L105 So SWAMPS is based on microwave satellite observations – what is the observational data that GLWD is based on?

L127 I didn't fully understand the scaling – are the mean/standard deviation global values?

L130 As previous comment - is the global mean 0?

L132 A modern-day test data set...

L134 conducted on -> driven by?

L136 Use the same terminology as I132 to avoid confusion, i.e.: "The paleoclimatic assessment of our model was performed using an early Eocene..." -> An early Eocene test data set was made using...?

L145 It would be useful to provide a summary table of the test/reference data sets to clarify exactly how you are going to evaluate your approach; I didn't find the current explanation completely clear.

L163 The number of what? Months or grid cells?

L214 Is Rh an absolute or scaled (0-1) value? If absolute, what are the units? Similarly for GPP in the next equation (I guess it must be absolute value to make sense.)

L218 Is TMP soil, surface, or surface air temperature?

L226 Presumably me >= 0? Is there a test for mp >= mo? What are the units of me?
L232 'downscaled' – I think the definition of downscaling is to infer something at high resolution from something at low resolution. You seem to be using the word in the opposite sense. I don't think we (scientists) normally use 'upscaled', so I am unsure what to call this (degrading?), but I don't think it is downscaling (also I234).

L318 I think the term 'maxKNN' appears here for the first time and isn't defined. Is it just KNN with K>1? (As suggested by I316.)

L364 In a similar vein to the last comment - why not just 3NN rather than max3NN?

L383 '...both [FW and CH4 emissions] have their highest values in summer months...' This is not so clear in Figure 8 for SDGVM. It is clear in Figure 9.

L409 '... their respective impacts of soil water balance...'. Clarify. Is this just a typo of -> on?

L409 I got a bit confused here about EVT. It seems EVT is from the vegetation models; but EVT must also be calculated in the underlying climate model – I guess with a much more simplified vegetation scheme. Is there a large discrepancy between the EVT in the vegetation and climate models? Isn't this a bit of a problem? This decoupling of the simulated water budget between the climate model and the vegetation model should be clearly explained earlier in the methods section, and the implications discussed here.

L423 Global monthly mean FW for the Eocene...

L572 Figure 1 caption – Annual monthly maximum...

L586 Figure 4 caption and y-axes – clarify these are CH4 emissions – what are the units? (Tg CH4/month?)

L628 Incorrect punctuation for list.

---

## Author Comment (AC2) · 18 Jan 2019

We thank the referee for their review and will revise the text in response to each of the points raised. We will make the minor alterations and clarifications the referee has requested in due course. Below we respond to the more substantive comments.

"I am fairly convinced this is a sensible and useful approach, but I must admit to being slightly baffled about the exact methods employed - I found the paper rather unclear in quite a few places. I would encourage the authors to revise the description of the methods to make it clearer. "

We will add to the introduction to better outline the paper and methods we explore.

"Some clarification on how this approach should be employed by the wider modelling

community would also be appreciated – can the method be embedded within ESMs to calculate wetland emissions online? Or is it envisaged as an offline only tool? I wasn't clear. If the authors can clarify the methods"

This has been used as an offline tool. We will add to the discussion at the end with respect to this.

"L71 So you are using DGVMs to simulate vegetation distributions, rather than using present-day observational datasets. It may be worth saying that the DGVMs have (presumably) been evaluated elsewhere."

We give references for the two DGVMs we use when they are first mentioned in section 2.1. Earlier, L71, we think it should be sufficient to simply say that references for those DGVMs will be given later in the text, and will revise L71 accordingly.

"L68 Is it worth briefly defining wetland? Perhaps earlier. E.g. the RAMSAR definition. Is it obvious how such definitions translate into a climate model-specific definition? (Water depth, etc.). What is the basis of the modern day reference data set of FW? Can you say it is 'known' or 'observed' FW? "

We will add to the text to briefly define wetlands. We use the term 'observed' to distinguish the reference FW from our later modelled FW. This will be clarified.

"L145 It would be useful to provide a summary table of the test/reference data sets to clarify exactly how you are going to evaluate your approach; I didn't find the current explanation completely clear."

We will add such a table.

"L318 I think the term 'maxKNN' appears here for the first time and isn't defined. Is it just KNN with K>1? (As suggested by I316.) "

Yes the term maxKNN does appear for the first time here in the section title. We will rename that section to "Maximum of K nearest neighbours FW prediction" and define

GMDD
maxKNN later in that section

"L364 In a similar vein to the last comment - why not just 3NN rather than max3NN?"

'3NN' does not indicate it is the maximum of those 3 nearest neighbours, it could imply any function of the three nearest neighbours, therefore we prefer to use max3NN.

"L409 I got a bit confused here about EVT. It seems EVT is from the vegetation models; but EVT must also be calculated in the underlying climate model – I guess with a much more simplified vegetation scheme. Is there a large discrepancy between the EVT in the vegetation and climate models? Isn't this a bit of a problem? This decoupling of the simulated water budget between the climate model and the vegetation model should be clearly explained earlier in the methods section, and the implications discussed here."

EVT, as used throughout the paper, is always from the vegetation models. We have not considered EVT from the climate model. So long as it is the same EVT used at all times in our modelling of FW, i.e. same definition for reference and test data sets, this should not be an issue. We will clarify that here.

---

## Author Response (AR1)

**1 A Predictive Algorithm For Wetlands In Deep Time Paleoclimate Models**

- 2 David J. Wilton1, Marcus Badger2,3,4, Euripides P. Kantzas1, Richard D. Pancost3, Paul J.
- 3 Valdes4, David J. Beerling1
- 4

[revised manuscript text omitted]